# Diverse Mechanisms of Resistance against Osimertinib, a Third-Generation EGFR-TKI, in Lung Adenocarcinoma Cells with an *EGFR*-Activating Mutation

**DOI:** 10.3390/cells11142201

**Published:** 2022-07-14

**Authors:** Shigetoshi Nishihara, Toshimitsu Yamaoka, Fumihiro Ishikawa, Tohru Ohmori, Koichi Ando, Sojiro Kusumoto, Yasunari Kishino, Ryo Manabe, Yuki Hasebe, Hironori Sagara, Hitoshi Yoshida, Junji Tsurutani

**Affiliations:** 1Division of Gastroenterology, Department of Medicine, Showa University School of Medicine, Tokyo 142-8666, Japan; s.nishi@med.showa-u.ac.jp (S.N.); hyoshida@med.showa-u.ac.jp (H.Y.); 2Advanced Cancer Translational Research Institute, Showa University, Tokyo 142-8555, Japan; y.hasebe@med.showa-u.ac.jp (Y.H.); tsurutaj@med.showa-u.ac.jp (J.T.); 3Division of Respirology and Allergology, Department of Medicine, Showa University School of Medicine, Tokyo 142-8666, Japan; ohmorit@med.showa-u.ac.jp (T.O.); koichi-a@med.showa-u.ac.jp (K.A.); k-sojiro@med.showa-u.ac.jp (S.K.); ookiyookiy@med.showa-u.ac.jp (Y.K.); r.manabe@med.showa-u.ac.jp (R.M.); sagarah@med.showa-u.ac.jp (H.S.); 4Center for Biotechnology, Showa University, Tokyo 142-8555, Japan; f-ishikawa@pharm.showa-u.ac.jp

**Keywords:** drug resistance, osimertinib, third-generation epidermal growth factor receptor tyrosine kinase inhibitor, KRAS, EGFR

## Abstract

Osimertinib, a third-generation epidermal growth factor receptor (EGFR) tyrosine kinase inhibitor (TKI), is used as a first-line treatment for patients with EGFR-mutant non-small cell lung cancer (NSCLC). However, the mechanisms underlying its anticancer activity, particularly the subsequent development of acquired resistance, are unclear. Herein, we investigated the mechanisms underlying the development of osimertinib resistance by treating NSCLC PC-9 cells (harboring an EGFR-activating mutation) with osimertinib, thereby developing five resistant cell lines, i.e., AZDR3, AZDR6, AZDR9, AZDR11, and AZDR14. The amplification of wild-type *EGFR* in AZDR3 cells and wild-type EGFR and KRAS in AZDR6 cells was also studied. AZDR3 cells showed dependence on EGFR signaling, in addition to afatinib sensitivity. AZDR9 cells harboring *KRAS^G13D^* showed sensitivity to MEK inhibitors. Furthermore, combination treatment with EGFR and IGF1R inhibitors resulted in attenuated cell proliferation and enhanced apoptosis. In AZDR11 cells, increased Bim expression could not induce apoptosis, but Bid cleavage was found to be essential for the same. A SHP2/T507K mutation was also identified in AZDR14 cells, and, when associated with GAB1, SHP2 could activate ERK1/2, whereas a SHP2 inhibitor, TNO155, disrupted this association, thereby inhibiting GAB1 activation. Thus, diverse osimertinib resistance mechanisms were identified, providing insights for developing novel therapeutic strategies for NSCLC.

## 1. Introduction

The identification of epidermal growth factor receptor (*EGFR*)-activating mutations has introduced a paradigm shift in the strategies used for the treatment of non-small cell lung cancer (NSCLC). *EGFR*-activating mutations, 90% of which comprise 15 bp deletions in exon 19 or L858R point mutation in exon 21, occur in nearly 50% of Asian patients and 10–15% of Caucasian patients with NSCLC [1,2].

EGFR tyrosine kinase inhibitors (TKIs) are the standard first-line treatment for locally advanced or metastatic EGFR-mutant NSCLC. However, despite significant benefits, most patients acquire resistance against these TKIs. The most common mechanisms underlying the development of this resistance involve the acquisition of the gatekeeper T790M mutation in EGFR exon 20 [3].

Osimertinib is an irreversible third-generation EGFR-TKI that targets the cysteine 797 residue in the ATP-binding site of EGFR kinase via covalent bond formation. It is highly selective for EGFR-activating mutations, as well as the *EGFR*-T790M mutation [4]. Patients with EGFR-T790M-positive NSCLC who were treated with osimertinib showed a higher response rate and prolonged progression-free survival (PFS) than those treated with platinum-based chemotherapy [5]. Moreover, compared with first-generation EGFR-TKIs, osimertinib exhibits higher efficacy and a lower toxicity profile, as well as longer PFS and overall survival in treatment-naïve patients with EGFR-mutant-advanced NSCLC [6,7]. Therefore, osimertinib is recognized as the first-line treatment for patients with EGFR-mutant NSCLC.

Although osimertinib plays an important role in NSCLC treatment, acquired resistance remains a significant obstacle, and the underlying mechanisms have not been completely elucidated. However, C797, G796, and L792 mutations in EGFR exon 20, amplification of MET and ERBB2, mutations in RAS and PIK3CA, small cell transformation, and epithelial-mesenchymal transition represent the possible drivers of resistance. Nevertheless, approximately 30–40% of the resistance-related mechanisms remain challenging to overcome [8]. Several new and emerging combinations of therapies can potentially improve the outcomes of EGFR-mutant NSCLC. With advancements in precision medicine in oncology, the selection of the correct treatment for resistant tumors will become increasingly important both before initial treatment and after disease progression.

Our group has previously reported on the mechanisms by which tumors acquire resistance against third-generation EGFR-TKIs (osimertinib and rociletinib) using afatinib-resistant lung adenocarcinoma PC-9 cells harboring an acquired T790M mutation in *EGFR* [9]. Herein, we elucidate the mechanisms by which five PC-9 cell lines harboring an EGFR-activating mutation, i.e., 15 bp deletion in EGFR exon 19, acquire resistance against osimertinib administered as a first-line treatment. Our findings may contribute to the development of novel therapeutic strategies for NSCLC patients with EGFR mutations.

## 2. Materials and Methods

### 2.1. Cell Lines and Reagents

The human lung adenocarcinoma cell line, PC-9, established from an untreated patient as previously described [10], was donated by K. Hayata. PC-9 cells, which were cultured in RPMI-1640 medium with 10% fetal bovine serum, penicillin (100 U/mL), and streptomycin (100 µg/mL) in a 5% CO_2_ incubator at 37 °C. The cells were passaged for less than 4 months before renewal from frozen stocks. The cell lines were authenticated using short tandem repeat analysis at the Japanese Collection of Research Bioresources Cell Bank (Promega, Madison, WI) and tested for *Mycoplasma* using a MycoAlert Mycoplasma detection kit (Lonza, Basel, Switzerland). Afatinib and xentuzumab were from Boehringer-Ingelheim (Ingelheim am Rhein, Germany); other inhibitors and chemicals were obtained from Selleck Chemicals (Houston, TX, USA) and Sigma-Aldrich (St. Louis, MO, USA), respectively.

### 2.2. Establishment of Acquired Osimertinib-Resistant PC-9 Cells Harboring a 15 bp Deletion in EGFR Exon 19 as Front-Line Therapy

PC-9 cells, established from untreated patients and harboring a 15 bp deletion in EGFR exon 19, were exposed to escalating concentrations of osimertinib in the growth medium in a stepwise manner. One-tenth of the half-maximal inhibitory concentration, i.e., 30 nM, was used as the starting concentration, and the concentration was gradually increased over a duration of 10–12 months to 1 µM osimertinib. After the osimertinib concentration reached 1 µM, these resistant cell lines were cultured in 1 µM osimertinib-containing medium for at least 2–3 months before testing. The resulting PC-9 osimertinib-resistant cell lines were named AZDR3, AZDR6, AZDR9, AZDR11, and AZDR14 and maintained in the growth culture medium containing 1 µM osimertinib (Figure 1a).

### 2.3. Cell Viability Assay

Cell proliferation and inhibition were measured using the MTT assay (Promega, Madison, WI, USA), as described previously [11]. Six to twelve replicates were prepared, and the experiments were repeated three times.

### 2.4. Immunoblotting, Immunoprecipitation, and Antibodies

Immunoblotting and immunoprecipitation were performed as described previously [9]. The primary antibodies used in these experiments are listed in Appendix A.

### 2.5. Subcellular Fractionation

Cytosolic and mitochondrial extracts were prepared using the EzSubcell Fraction from ATTO (Tokyo, Japan), in accordance with the manufacturer’s instructions. The proteins in each fraction were solubilized and analyzed via immunoblotting.

### 2.6. RAS Pull-Down Assay

The activation of RAS or KRAS was measured using pull-down assays, following the manufacturer’s instructions (EMD Millipore, Temecula, CA, USA). A glutathione S-transferase fusion protein was used, which corresponds to the human RAS-binding domain of RAF-1 and binds to the GTP-bound form of RAS. Western blots were developed using anti-RAS or anti-KRAS antibodies.

### 2.7. RNA Interference

Non-targeting (N/T) short interfering RNA (siRNA) (controls) and SMARTpool siRNAs targeting EGFR (M003114) and PTPN11 (M003947) were purchased from Dharmacon (Lafayette, CO, USA). Briefly, 1 × 10^5^ cells/well in 6-well plates were transfected with siRNAs (100 pmol siRNA/well) using Lipofectamine 2000 reagent (Life Technologies, Carlsbad, CA, USA) and following the manufacturer’s instructions.

### 2.8. Quantitative PCR

RT-PCR was performed using aliquots of cDNA or genomic DNA (gDNA), as described previously [9]. gDNA and RNA extractions were performed using the QIAamp DNA Mini Kit (Qiagen, Valencia, CA) and RNeasy Plus Mini Kit (Qiagen), respectively, and cDNA was synthesized using the TaKaRa PrimeScript RT reagent kit (Takara Bio, Kusatsu, Japan). The expression of mRNA and gDNA was normalized to that of GAPDH or LINE1, respectively. Specific primer sets are listed in Appendix A [10,12].

### 2.9. EGFR, KRAS, and PTPN11 Sequence Analysis

Exon 19 of EGFR, exon 2 of KRAS, and exon 13 of PTPN11 were amplified using TaKaRa ExTaq polymerase (Takara Bio, Otsu, Japan) from gDNA with specific PCR primers, according to the manufacturer’s instructions. The products were purified and sequenced by FASMAC (Atsugi, Japan). The primers used for the PCR and sequencing are listed in Appendix A [10,12].

### 2.10. Droplet Digital PCR Analysis for EGFR Allele Quantification

An LBx probe for EGFR exon 19 del was used for droplet digital PCR, and gDNA samples were quantified using a QX200 droplet reader (Bio-Rad, Hercules, CA, USA) at Riken Genesis (Kawasaki, Japan).

### 2.11. PCR Analysis of EGFR Exon 19

EGFR was amplified using TaKaRa ExTaq polymerase (Takara Bio, Kusatsu, Japan), with previously described primers [13], and we analyzed the wild-type (WT) EGFR expression and 15 bp deletion in exon 19.

### 2.12. Xenograft Studies on Tumor Growth

Six-eight-week-old female severe combined immunodeficiency (SCID) mice (strain: C.B-17/IcrHsd-Prkdcscid) were purchased from Sankyo Labo Service Corporation (Tokyo, Japan), housed, and treated according to institutional guidelines. Cell suspensions (5 × 10^6^ cells/mice) were injected subcutaneously into the flank of each SCID mouse. The mice were randomized (*n* = 6) after the mean tumor volume reached approximately 250 mm^3^. Osimertinib (5 mg/kg/day) and afatinib (6 mg/kg/day) were suspended in 0.5% 2-hydroxyethyl cellulose/H_2_O and administered orally once daily. The tumors were measured twice weekly, and xenograft tumor volumes were calculated using the following formula: length^2^ × width × 0.5.

### 2.13. Cell-Based Insulin-like Growth Factor 1 Receptor (IGF1R) Phosphorylation Assay

IGF bioactivity was determined using mouse embryonic fibroblasts (MEFs) derived from IGF1R-deficient mice. Human IGF1R was overexpressed in MEFs (MEF-IGF1R), as described previously [10]. IGF1 (80 ng/mL) was used as a positive control. A culture medium of PC-9, AZDR11, or AZDR14 cells was used.

### 2.14. Next-Generation Sequencing and nCounter Analysis

To investigate resistance mechanisms, PC-9 and AZDR14 cells were tested using the next-generation sequencing OncoGxOne panel at GENEWIZ (Saitama, Japan). To analyze mRNA expression, PC-9, AZDR3, and AZDR6 cells were tested using the nCounter Pancancer Pathways Panel by AS ONE (Tokyo, Japan).

### 2.15. Caspase-3/7 Activity Assay

To determine the induction of apoptosis, the activity of caspase-3/7 was measured using the Caspase-Glo 3/7 assay kit (Promega), in accordance with the manufacturer’s instructions.

### 2.16. Statistical Analysis

Data are presented as the mean ± standard error of the mean (SEM) and were analyzed using GraphPad Prism version 9.3 (GraphPad, Inc., San Diego, CA, USA). Significance was evaluated using two-tailed Student’s *t*-tests, and *p* < 0.05 was considered significant.

## 3. Results

### 3.1. Characteristics of Clones with Acquired Resistance against Osimertinib

Cell proliferation was effectively suppressed in PC-9 cells harboring the 15 bp deletion in exon 19 (exon 19 del) of EGFR by osimertinib, but not in the established osimertinib-resistant AZDR cell lines (Figure 1b). Notably, AZDR3, AZDR6, and AZDR11 cells overexpressed EGFR, whereas the basal expression of EGFR exon 19 del did not change, compared to that in PC-9 cells (Figure 1c). The expression and activation of other receptor tyrosine kinases did not vary (Figure 1c and Appendix A), while that of WT EGFR increased in these cell lines. Additionally, EGFR was significantly amplified (*p* < 0.01) (Figure 1d,e and Appendix A), and the homoduplexes of WT EGFR increased in the AZDR3, AZDR6, and AZDR11 cells (Figure 1e). Digital PCR revealed that the copy number of EGFR exon 19 del, relative to WT, was lower in the AZDR3, AZDR6, and AZDR11 cells than in other cells, indicating that WT EGFR amplification is required for resistance (Figure 1f). As third-generation EGFR-TKIs are relatively selective inhibitors of the aberrant activation of mutant EGFR, the increased WT EGFR expression represents one potential mechanism underlying the acquisition of osimertinib resistance. These resistant cell lines, exhibiting increased homoduplexes of WT EGFR, showed relatively slower cell proliferation than PC-9 and other resistant cell lines (Figure 1g). Conversely, AZDR9 and AZDR14 cells proliferated faster than AZDR3, AZDR6, and AZDR11 cells (Figure 1g).

### 3.2. AZDR3 Cells Exhibit EGFR Amplification and Loss of EGFR Exon 19 Deletion

As WT EGFR was amplified in AZDR3 cells, its transcript-level expression was higher than that of other molecules determined using the nCounter screening analysis system (Appendix A). Moreover, the homoduplex of WT EGFR was predominant (Figure 1e); the suppressive effect of osimertinib on EGFR phosphorylation was minimal (Figure 2a), although osimertinib clearly inhibited EGFR phosphorylation in PC-9 cells in a concentration-dependent manner (Figure 2b). Afatinib, which irreversibly inhibits the phosphorylation of mutated EGFR and WT EGFR, suppressed cell proliferation in AZDR3 cells (Figure 2c) and inhibited EGFR phosphorylation and downstream AKT and ERK1/2 activation, followed by apoptosis induction (Figure 2d). Furthermore, siRNA-mediated EGFR knockdown inhibited AKT and ERK1/2 activation and led to apoptosis in AZDR3 cells, similar to that observed in PC-9 cells, suggesting that amplified WT EGFR expression is necessary for osimertinib resistance (Figure 2e). Although overexpression of EGFR ligands is associated with resistance against EGFR-TKIs [14], the ligands were not upregulated in AZDR3 cells (Appendix A).

Next, PC-9 and AZDR3 cells were transplanted into SCID mice, followed by oral administration of osimertinib or afatinib. The AZDR3 xenograft tumors were not suppressed by osimertinib administration, whereas the PC-9 xenograft tumors regressed completely. Additionally, afatinib treatment resulted in regressed tumor growth in the PC-9 and AZDR3 xenografts (compared to the control) (Figure 2f). Taken together, irrespective of EGFR ligand overexpression, WT EGFR amplification led to excessive EGFR phosphorylation, resulting in the development of osimertinib resistance.

### 3.3. Increase in Acquired WT KRAS Expression in AZDR6 Cells, and KRASG13D Mutation in AZDR9 Cells Results in Differential Sensitivity to MEK Inhibitor

Basal phosphorylation of ERK1/2 was higher in AZDR6 and AZDR9 cells than in the PC-9 and other AZDR cell lines (Figure 1c). Therefore, the expression of RAS was detected by immunoblotting and compared with that of the parental PC-9 cells. KRAS expression and activation were higher in AZDR6 and AZDR9 cells than in PC-9 cells (Figure 3a). Hence, exons 2–4 of KRAS were analyzed by direct sequencing, and the KRAS^G13D^ mutation was identified in KRAS exon 2 in the AZDR9 cells; no KRAS mutations were detected in the AZDR6 cells (Figure 3b). KRAS amplification or mutations led to the constitutive activation of the RAS-MEK pathway. Therefore, the sensitivity to MEK inhibitors in the presence or absence of EGFR-TKIs was determined in AZDR6 and AZDR9 cells (Figure 3c,d). Proliferation of AZDR6 cells overexpressing WT KRAS was inhibited by selumetinib, a MEK inhibitor, in the presence of afatinib; however, selumetinib or afatinib alone did not inhibit cell proliferation. Selumetinib inhibited ERK1/2 phosphorylation, but not AKT phosphorylation. When afatinib was combined with selumetinib, AKT and ERK phosphorylation was completely inhibited, leading to induction of apoptosis in AZDR6 cells (Figure 3e). AZDR9 cell proliferation was inhibited by selumetinib, irrespective of osimertinib treatment. Therefore, selumetinib inhibited both AKT and ERK1/2 phosphorylation in a concentration-dependent manner (Figure 3f). The basal EGFR phosphorylation level was as high in AZDR6 cells as in PC-9 cells, but it was substantially suppressed in the AZDR9 cells, indicating that the emergence of KRAS^G13D^ completely abolished EGFR signaling. Interestingly, the MEK inhibitor could inhibit the phosphorylation of ERK1/2 and AKT. This effect was observed not only with selumetinib, but also with trametinib, another MEK inhibitor, which reduced both AKT and ERK1/2 activation (Appendix A). Further investigation is required to clarify whether KRAS^G13D^ mediates specific effects.

Overexpression of WT KRAS was reportedly attenuated in parental PC-9 cells upon culturing them in EGFR-TKI-free conditions for 2 months [9]. Similarly, WT KRAS expression and amplification decreased in AZDR6 cells; however, EGFR amplification was not affected (Appendix A). nCounter analysis revealed that the transcript-level expression of KRAS increased independently in AZDR6 cells, compared to that in PC-9 cells (Appendix A). AZDR6-F2M cells cultured in osimertinib-free media for 2 months exhibited a reduced transcript-level expression of KRAS (Appendix A). This resulted in recovered sensitivity to afatinib and Osimertinib, both in vitro and in vivo (Appendix A). Furthermore, afatinib or osimertinib inhibited EGFR phosphorylation in a concentration-dependent manner; the downstream signaling of AKT and ERK1/2 was also inhibited, leading to apoptosis (Appendix A). Taken together, these results suggest that KRAS mutations and WT KRAS amplification confer osimertinib resistance; when this amplification is attenuated, resistance against EGFR-TKIs is reversed.

### 3.4. Bypass Signal of IGF1R to AKT Requires Ligand(s) Stimulation in AZDR11 and AZDR14 Cells

The bypass signal of IGF1R is reportedly a survival pathway for escaping the inhibition of EGFR activity by EGFR-TKIs [15,16]. In AZDR11 and AZDR14 cells, osimertinib treatment did not suppress IGF1R phosphorylation; however, EGFR phosphorylation was attenuated (Figure 4a). In AZDR11 cells, osimertinib inhibited ERK1/2 activation, suggesting that the ERK1/2 pathway is regulated by EGFR (Figure 4a). When linsitinib, an IGF1R inhibitor, was used along with osimertinib, AZDR11 cell proliferation was suppressed, whereas AZDR14 cell proliferation was not (Figure 4b). In AZDR11 cells, osimertinib inhibited ERK1/2 activation, linsitinib inhibited AKT activation, and combination treatment induced apoptosis (Figure 4c). Similarly, afatinib inhibited ERK1/2 activation in a concentration-dependent manner, while linsitinib inhibited AKT activation in AZDR11 cells (Appendix A). In contrast, in AZDR14 cells, IGF1R and AKT activation was inhibited by linsitinib, suggesting that IGF1R regulates the AKT pathway. However, the activation of ERK1/2 was not affected, even by combination treatment with EGFR-TKI and IGF1R inhibitors (Figure 4c).

Next, to determine the mechanism underlying IGF1R activation in AZDR11 and AZDR14 cells, we compared the bioactivities of IGF1R in the culture medium of these cells to those of PC-9 cells using a cell-based IGF1R phosphorylation assay, in which IGF1R phosphorylation was quantified using ELISA [17]. The phosphorylated IGF1R level was higher in the culture media of AZDR11 and AZDR14 cells than in that of PC-9 cells (Figure 4d). When the humanized monoclonal antibody, xentuzumab, neutralized the IGF1R ligands of IGF1 and IGF2 [17], IGF1R phosphorylation and AKT activation were attenuated concentration-dependently in AZDR11 and AZDR14 cells (Figure 4e and Appendix A). Furthermore, combination treatment with xentuzumab and osimertinib induced apoptosis and inhibited AZDR11 cell proliferation (Figure 4f,g). Therefore, in AZDR11 and AZDR14 cells, IGF1R activation regulates AKT activation as a bypass signaling pathway via ligand stimulation.

### 3.5. Bid Cleavage by Caspase-8 Is Required for Apoptosis Induction in AZDR11 Cells

To investigate the effect of osimertinib and/or linsitinib treatment on apoptosis in AZDR11 and PC-9 cells, we analyzed the expression of the anti-apoptotic proteins, Bcl-2, Bcl-xL, and Mcl-1, as well as pro-apoptotic proteins, Bim, Bad, Puma, Bid, Bax, and Bak (Figure 5a). Osimertinib treatment considerably altered the levels of Bad, phospho-Bad (S136 and S112), and Bim. One of the pro-apoptotic BH3-only proteins, Bad, is regulated by phosphorylation of serine 112 and serine 136 residues. Phosphorylation of either site results in the loss of the Bad function [18]. Osimertinib treatment increased Bad and phospho-Bad (S136) levels in PC-9 and AZDR11 cells, whereas phospho-Bad (S112) levels decreased when AZDR11 cells were treated with linsitinib (Figure 5a).

Bim upregulation was observed following treatment with osimertinib; however, apoptosis induction, represented by cleavage of caspase-3 and PARP, was only observed in PC-9 cells. In AZDR11 cells, cleavage of caspase-3 and PARP was detected after combination treatment with osimertinib and linsitinib. Bid cleavage was detected when AZDR11 cells were treated with osimertinib and linsitinib and PC-9 cells were treated with osimertinib (Figure 5a). Cleavage of Bid was consistent with apoptosis induction.

In the mitochondrial apoptosis pathway, Bax and Bak undergo conformational changes to their active forms, thus disrupting the mitochondrial membrane to release cytochrome c and leading to cleavage of caspase-3 [19]. Bax was immunoprecipitated with 6A7 antibody, which specifically binds to the active form of Bax [20]. Activated Bax levels increased considerably in PC-9 cells treated with osimertinib and AZDR11 cells treated with a combination of both osimertinib and linsitinib (Figure 5b). Bax activation was consistent with the increase in the levels of cleaved Bid (Figure 5a). Cytochrome c levels in the cytoplasm also increased (Figure 5b). The increase in caspase-3/7 activity was consistent with the levels of cleaved Bid and activated Bax (Figure 5c). These results indicated that the increase in cleaved Bid after combination treatment of AZDR11 cells with osimertinib and linsitinib may be associated with the activation of the mitochondrial apoptotic pathway via activation of Bax, thus leading to cytochrome c release into the cytoplasm and caspase-3 activation. Cleaved Bid translocates to the mitochondria and induces mitochondrial apoptosis [21]. As shown in Figure 5d, treatment of PC-9 cells with osimertinib and combination treatment of AZDR11 cells with osimertinib and linsitinib led to the translocation of cleaved Bid to the mitochondria. Conversely, a Bid-specific inhibitor, BI-6C9, inhibited cleavage of Bid and caspase-3 in PC-9 and AZDR11 cells (Figure 5e). BI-6C9 treatment suppressed the increased caspase-3/7 activity after the exposure of PC-9 cells to osimertinib and AZDR11 cells to a combination of osimertinib and linsitinib (Figure 5f). These results suggest that apoptotic signaling is transduced via Bid/Bax in mitochondria, and Bid cleavage is required for the induction of apoptosis in PC-9 and AZDR11 cells.

Cleavage of Bid can be induced by activated caspase-8, which can also directly cleave caspase-3 [22]. Activated caspase-8 was detected in the PC-9 cells exposed to osimertinib and AZDR11 cells exposed to osimertinib and linsitinib (Figure 5g). To assess the effect of caspase-8 activation on Bid, PC-9, and AZDR11 cells were treated with a caspase-8 inhibitor, z-IETD-fmk, in the presence or absence of osimertinib and/or linsitinib. Treatment with z-IETD-fmk considerably abrogated caspase-8, and Bid cleavage induced by osimertinib in PC-9 cells and combination treatment with osimertinib and linsitinib in AZDR11 cells (Figure 5h). The effect of the caspase-8 inhibitor on Bid cleavage suggested the involvement of the extrinsic apoptotic pathway. The mitochondrial pathway involving caspase-8 activation and the cleavage of Bid transduced the extrinsic apoptotic signaling to the mitochondrial apoptosis signaling pathway.

### 3.6. An Acquired Novel Mutation, T507K PTPN11 Detected in AZDR14 Cells Activates ERK1/2 Signal via Association with SHP2/GAB1

In AZDR14 cells, ERK1/2 activation was not blocked by the EGFR inhibitor and/or IGF1R inhibitor (Figure 4c), although AKT activation was inhibited by IGF1R inhibitors linsitinib or xentuzumab (Appendix A). Total RAS activity was upregulated, compared to that in the parental PC-9 cells; however, total RAS expression levels were not affected (Figure 6a). Next-generation sequencing was performed to identify the mutation(s) that can upregulate the RAS-ERK1/2 signaling pathway. A 1520C>A mutation in exon 13 (T507K) of PTNPN11, encoding the non-receptor tyrosine phosphatase, SHP2, was identified and confirmed via direct sequence analysis (Figure 6b). SHP2 plays an important role in regulating cellular events downstream of various growth factors, cytokines, and integrin receptors [23]. The transforming potential of SHP2/T507K is associated with aberrant RAS activation and high affinity for the scaffolding protein GAB1 [24]. To determine the effect of SHP2/T507K on AZDR14 cells, PTPN11 siRNA was transfected with or without osimertinib and/or linsitinib treatment. The basal levels of GAB1 and ERK1/2 phosphorylation decreased when PTPN11 was knocked down. The combination of osimertinib and linsitinib considerably inhibited AKT and ERK1/2 activation, and the induction of apoptosis and suppression of cell proliferation were observed following transfection of PTPN11 siRNA (Figure 6c,d).

The effect of an allosteric SHP2 inhibitor, TNO155, on cell proliferation was assessed in the presence or absence of osimertinib and/or linsitinib. TNO155 considerably suppressed AZDR14 cell proliferation in the presence of osimertinib and linsitinib (Figure 6e). In AZDR14 cells, TNO155 inhibited GAB1 phosphorylation; however, ERK1/2 and AKT activation was modestly inhibited. Moreover, in combination with osimertinib and linsitinib, TNO155 inhibited AKT and ERK1/2 activation, leading to apoptosis induction (Figure 6f). We investigated the association of SHP2/T507K with GAB1, a molecular event essential for SHP2-mediated RAS activation [25]. Osimertinib disrupted the association of SHP2 and GAB1 in PC-9 cells; however, in AZDR14 cells, this association was not inhibited, even after treatment with osimertinib and/or linsitinib (Figure 6g,h). TNO155 completely inhibited the association between SHP2 and GAB1 when cells were treated with a combination of osimertinib and linsitinib (Figure 6h). Taken together, these results suggest that the SHP2/T507K mutant binds strongly to GAB1, and enhanced engagement of SHP2/T507K-GAB1 is essential for excessive activation of the RAS-ERK1/2 pathway.

## 4. Discussion

In the present study, we determined five mechanisms of acquired resistance against the irreversible mutation-selective EGFR-TKI, osimertinib, a first-line therapy. First, enhanced WT EGFR expression in AZDR3 cells, without ligand overexpression, increased the homoduplexes of WT EGFR and resistance against osimertinib. Second, the observed WT KRAS and EGFR amplification in AZDR6 cells, as has been previously reported for afatinib resistance [12] and osimertinib resistance in PC-9 cells harboring T790M-EGFR [9], decreased within 2 months of cultivation in osimertinib-free medium. Moreover, these cells were re-sensitized to osimertinib after this period. Third, KRASG13D mutation was detected in AZDR9 cells, where the signals between EGFR and the downstream molecules ERK1/2 and AKT were perturbed. The MEK inhibitor selumetinib effectively inhibited cell proliferation. Fourth, the IGF1R bypass signaling pathway was observed, along with the induction of ligands in AZDR11 and AZDR14 cells. Interestingly, in AZDR11 cells, IGF1R primarily regulated AKT activation, whereas EGFR regulated ERK1/2 activation. Lastly, the SHP2/T507K mutation identified in AZDR14 cells activates ERK1/2 via association with the GAB1 complex. In AZDR11 cells, Bim upregulation was necessary, but not sufficient, for apoptosis induction. Bid cleavage was consistent with apoptosis induction. Our results provide evidence regarding heterogeneity in osimertinib resistance and reflect clinical resistance.

KRAS amplification and mutations are reportedly responsible for osimertinib resistance [26,27,28]. Recently, Roper et al. performed multi-region whole-exome and RNA sequencing of pre- and post-resistant tumors from patients with EGFR-mutant lung cancer treated with osimertinib. Among the 34 enrolled patients, one patient exhibited focal EGFR and KRAS amplification, which occurred concurrently in progressive tumor cells [27]. This case is similar to that of AZDR6 cells, which exhibited WT EGFR and WT KRAS amplification. Interestingly, WT KRAS amplification in AZDR6 cells was attenuated after 2 months of culture in osimertinib-free medium, although WT EGFR amplification was not attenuated, suggesting that a re-challenge with afatinib could effectively suppress cell proliferation. Taken together, these results suggest that multiple focal amplifications occur spatially and temporally during the acquisition of osimertinib resistance.

Bim elevation plays a critical role in the mechanism of action of EGFR-TKIs and induces apoptosis in NSCLC cells with EGFR-activating mutations [29,30,31]. Here, we report, for the first time, that Bim elevation is not sufficient for apoptosis induction in osimertinib-resistant AZDR11 cells. Moreover, cleavage of Bid, its mitochondrial translocation, and Bax activation were detected in both PC-9 and AZDR11 cells when apoptosis was induced. Bid plays a central role by connecting the extrinsic apoptotic pathway to the mitochondrial pathway [32]. Bid cleavage is mediated by activated caspase-8 via the extrinsic apoptotic pathway [33]. Osimertinib activates caspase-8 via cellular FLICE-inhibitory protein degradation and enhances the apoptosis induced by the tumor necrosis factor-related apoptosis-inducing ligand [34]. In PC-9 and AZDR11 cells, the inhibition of activated caspase-8 by z-IETD-fmk inhibited Bid cleavage. Therefore, Bid may be cleaved following caspase-8 activation, and it may be involved in the extrinsic apoptotic pathway in EGFR-TKI-induced apoptosis. Whether there is a connection between Bim elevation and Bid cleavage prior to the induction of apoptosis is unclear; this requires further investigation.

In the present study, SHP2/T507K mutation was associated with osimertinib resistance. This substitution has been reported in cases of neuroblastoma [35], hepatocellular carcinoma [36], and glioblastoma [37]. The T507K mutation in SHP2 can confer oncogenic RAS-like transforming potential to SHP2 [36]. SHP2 critically mediates the signaling of multiple receptor tyrosine kinases via the interaction of the phospho-tyrosine and SH2 domains. SHP2 can be activated by binding to phosphorylated GAB1 [38], while the GAB1-SHP2 complex acts upstream of RAS [39]. RAS activity was upregulated in AZDR14 cells, compared to that in PC-9 cells. The SHP2 inhibitor, TNO155 [40], was used for the dissociation of the GAB1-SHP2 interaction. TNO155 is currently being developed for clinical use for various types of cancers. In the present study, TNO155 dissociated the GAB1-SHP2/T507K complex and inhibited GAB1 phosphorylation; therefore, it would be beneficial for patients with SHP2/T507K mutation.

## 5. Conclusions

Our study clarified the novel mechanisms related to osimertinib resistance in lung adenocarcinoma PC-9 cells. Five osimertinib-resistant cells, i.e., AZDR3, AZDR6, AZDR9, AZDR11, and AZDR14, exhibiting different mechanisms of drug resistance were generated, highlighting the diverse nature of tumor evolution in osimertinib-resistant patients. The acquired resistance mechanisms reported in this study were obtained from a single cell line, PC-9. Therefore, it may be impossible to cover all of the mechanisms conferring resistance to osimertinib. Further, it was difficult to identify whether novel genetic alterations, bypass signals, or transformation confer resistance to osimertinib in human tissues. Therefore, preclinical models are required for understanding the resistance mechanisms. Further preclinical studies using NSCLC cell lines with activating EGFR mutations are required to identify novel resistance mechanisms to EGFR-TKI, as well as therapeutics to overcome the resistance. Our findings provide insights regarding future therapeutics for patients with NSCLC harboring EGFR-activating mutations.

## Figures and Tables

**Figure 1 cells-11-02201-f001:**
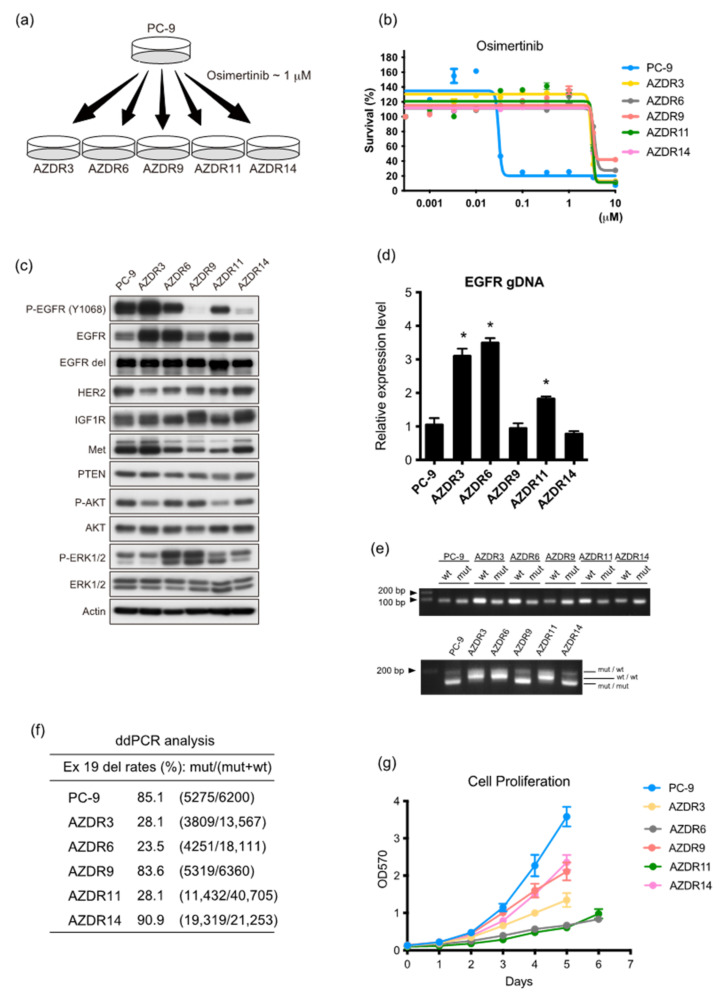
Establishment of osimertinib-resistant cell lines from PC-9 cells. (**a**) Schematic showing the process of generating osimertinib-resistant cell lines. (**b**) Cell viability, following osimertinib treatment for 72 h (n = 6). Data are presented as the mean ± standard error of the mean (SEM). (**c**) Western blot analysis of PC-9, AZDR3, AZDR6, AZDR9, AZDR11, and AZDR14 cells. β-Actin was used as the loading control. (**d**) *EGFR* expression in PC-9, AZDR3, AZDR6, AZDR9, AZDR11, and AZDR14 cells; * *p* < 0.01. (**e**) Upper: detection of wild-type (WT) and mutant (mut) epidermal growth factor receptor (*EGFR)* sequences using specific primers; lower: proportions of *EGFR* homoduplexes (mut/mut and WT/WT) and heteroduplexes (mut/wild-type). (**f**) Proportion of *EGFR* exon 19 del allele. (**g**) Cell proliferation analysis. OD_570_ values were measured on the indicated days (*n* = 6).

**Figure 2 cells-11-02201-f002:**
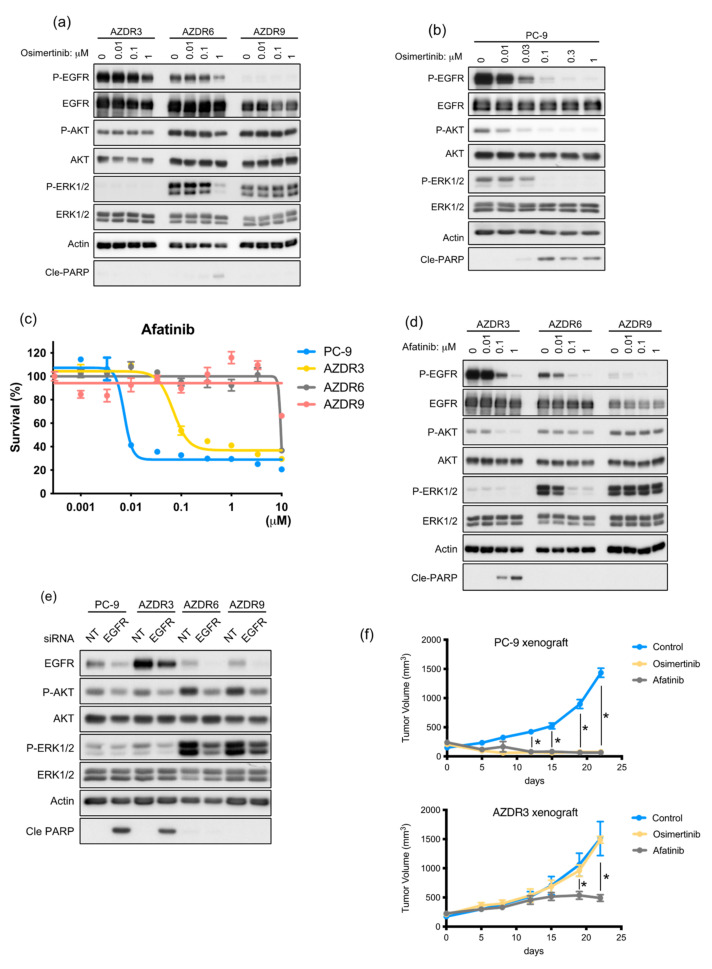
AZDR3 cells exhibit EGFR amplification and loss of EGFR exon 19 deletion, resulting in osimertinib resistance. (**a**,**b**) Western blot analysis of AZDR3, AZDR6, AZDR9 (**a**), and PC-9 (**b**) cells, following exposure to osimertinib for 24 h. (**c**) Cell viability following afatinib treatment for 72 h (*n* = 6). (**d**) Western blot analysis of AZDR3, AZDR6, and AZDR9 cells, following treatment with afatinib for 24 h. (**e**) EGFR knockdown in AZDR3, AZDR6, and AZDR9 cells, following transfection with non-targeting (N/T) siRNA or siRNA directed against EGFR. (**f**) Dimensions of tumors from female severe combined immunodeficiency mice injected with PC-9 and AZDR3 cells (*n* = 6) and treated with or without osimertinib (5 mg/kg) or afatinib (6 mg/kg). Data are presented as the mean tumor volume ± SEM; * *p* < 0.01. Cle-PARP, cleaved poly (ADP-ribose) polymerase.

**Figure 3 cells-11-02201-f003:**
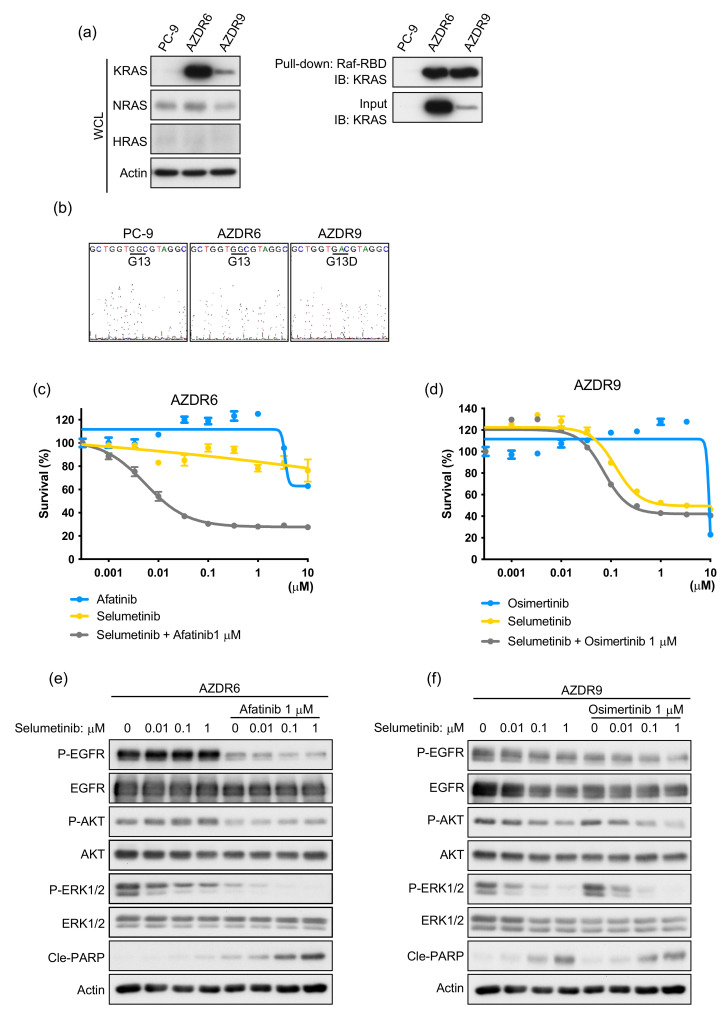
Increase in acquired wild-type KRAS expression in AZDR6 and KRASG13D mutations in AZDR9 cells results in differential sensitivity to MEK inhibitor. (**a**) KRAS, NRAS, and HRAS expression. (**b**) DNA sequence reads in KRAS exon 2. (**c**,**d**) Viability of AZDR6 (**c**) and AZDR9 (**d**) cells, following treatment with afatinib (**c**)/osimertinib (**d**) or selumetinib in the presence or absence of 1 μM afatinib (**c**)/osimertinib (**d**) for 72 h (*n* = 6). (**e**,**f**) Western blot analysis of AZDR6 and AZDR9 cells, following treatment with the indicated concentration of selumetinib in the presence or absence of afatinib (1 μM) (**e**) or osimertinib (1 μM) (**f**) for 24 h. Cle-PARP, cleaved poly (ADP-ribose) polymerase; WCL, whole-cell lysates.

**Figure 4 cells-11-02201-f004:**
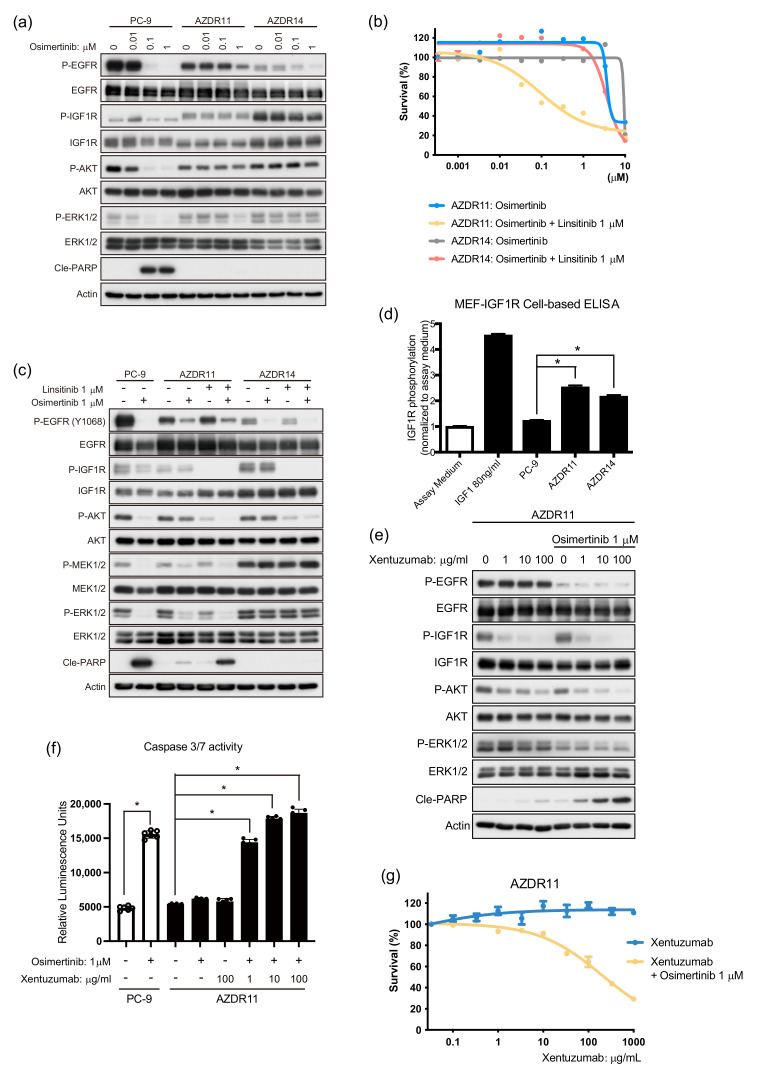
Bypass signal of insulin-like growth factor 1 receptor (IGF1R) to AKT requires ligand stimulation in AZDR11 and AZDR14 cells. (**a**) Western blot analysis of PC-9, AZDR11, and AZDR14 cells, following treatment with osimertinib for 24 h. (**b**) Viability of AZDR11 and AZDR14 cells, following treatment with osimertinib in the presence or absence of linsitinib (1 μM) for 72 h. (**c**) Western blot analysis of PC-9, AZDR11, and AZDR14 cells, following treatment with osimertinib for 24 h in the presence or absence of 1 μM linsitinib. (**d**) IGF1R phosphorylation in MEF-IGF1R cells treated with the conditioned culture media from PC-9, AZDR11, and AZDR14 cells. Data are represented as the mean ± SEM; * *p* < 0.01. (**e**) Western blot analysis of AZDR11 cells treated with xentuzumab for 24 h in the presence or absence of 1 μM osimertinib. (**f**) Caspase-3/7 activity in PC-9 and AZDR11 cells, following treatment with osimertinib (1 μM) in the presence or absence of xentuzumab for 24 h (*n* = 6). Data are represented as the mean ± SEM; * *p* < 0.01. (**g**) Proliferation of AZDR11 cells treated with xentuzumab in the presence or absence of osimertinib (1 μM) for 72 h. Cle-PARP, cleaved poly (ADP-ribose) polymerase.

**Figure 5 cells-11-02201-f005:**
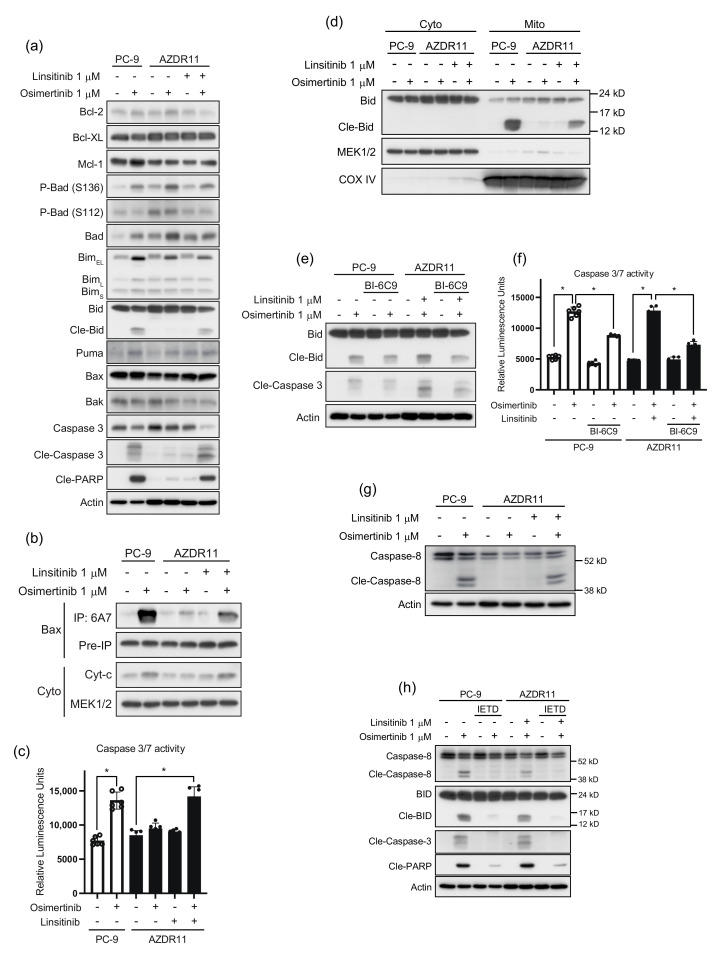
Bid cleavage by caspase-8 is required for induction of apoptosis in AZDR11 cells. (**a**) Effect of osimertinib and/or linsitinib treatment on the expression of pro-apoptotic and anti-apoptotic proteins. (**b**) Levels of active and total Bax and cytoplasmic (cyto) cytochrome c (cyt-c), following 24 h treatment with osimertinib (1 μM) and/or linsitinib (1 μM). (**c**) Caspase-3/7 activity in PC-9 and AZDR11 cells treated with osimertinib (1 μM) in the presence or absence of linsitinib (1 μM) for 24 h (*n* = 6). Data are presented as the mean ± SEM; * *p* < 0.01. (**d**) Expression of Bid and cleaved Bid in cytosolic (cyto) and mitochondrial (Mito) fractions of cells treated with osimertinib and/or linsitinib. MEK1/2 and Cox IV were used as internal controls. (**e**) Effect of BI-6C9 on Bid cleavage and caspase-3 activation. (**f**) Caspase-3/7 activity in cells treated with osimertinib (1 μM) or combined osimertinib (1 μM) and linsitinib (1 μM) for 24 h, with or without BI-6C9 (10 μM) treatment for 24 h. Data are presented as the mean ± SEM; * *p* < 0.01. (**g**) Caspase-8 activation. (**h**) Effect of caspase-8 inhibitor, z-IETD-fmk, on Bid cleavage, caspase-8 activation, and PARP cleavage. Cle, cleaved.

**Figure 6 cells-11-02201-f006:**
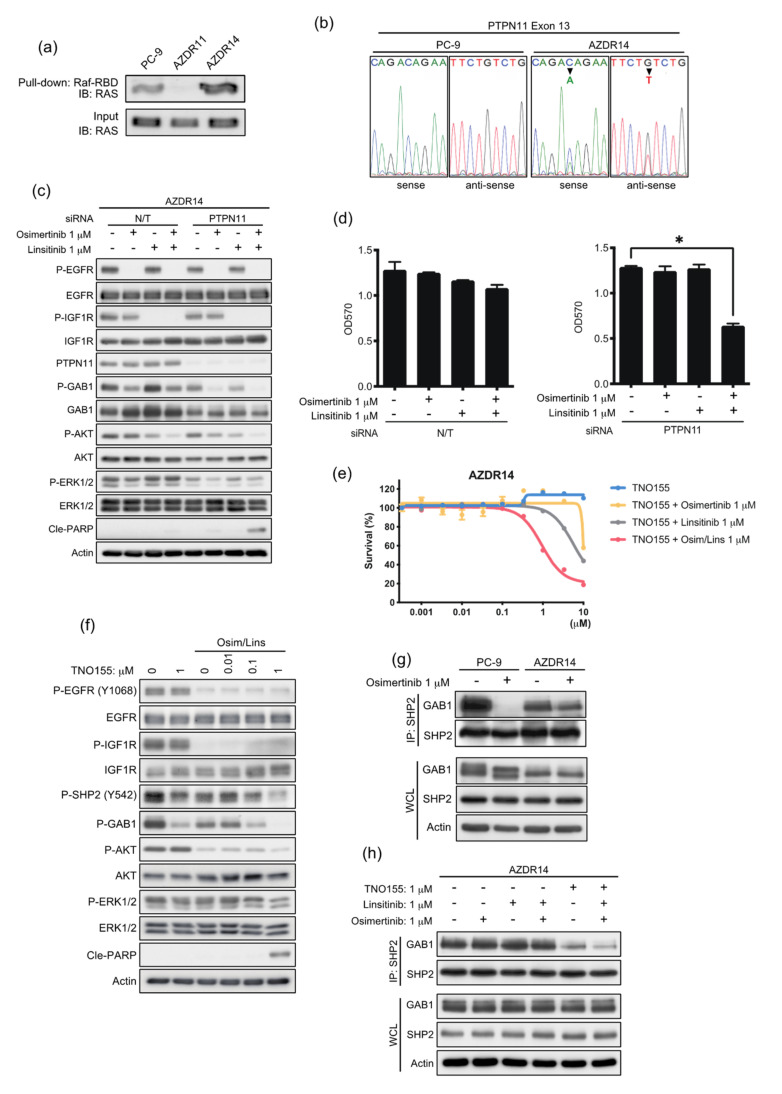
AZDR14 cells carry a novel T507K PTPN11 mutation that activates the ERK1/2 signal via association with SHP2/GAB1. (**a**) Active RAS in PC-9, AZDR11, and AZDR14 cells. (**b**) DNA sequence reads in PTPN11 exon 13. (**c**) PTPN11 knockdown in AZDR14 cells. (**d**) Proliferation of AZDR14 cells transfected with N/T siRNA or siRNA against PTPN11 and treated with or without 1 µM osimertinib and/or linsitinib (*n* = 6). Data are presented as the mean ± SEM; * *p* < 0.01. (**e**) Proliferation of AZDR14 cells, following treatment with TNO155 in the presence or absence of osimertinib (1 μM) and/or linsitinib (1 μM) for 72 h (*n* = 6). Data are presented as the mean ± SEM. (**f**) Western blot analysis of AZDR14 cells treated with TNO155 for 24 h in the presence or absence of combined osimertinib (Osim, 1 μM) and linsitinib (Lins, 1 μM). (**g**) Western blot analysis of whole-cell lysates (WCL) and SHP2-immunoprecipitates (IPs) from PC-9 and AZDR14 cells treated with or without osimertinib (1 μM) for 24 h. (**h**) Western blot analysis of WCL and SHP2 IPs from AZDR14 cells treated with osimertinib (1 μM) and/or linsitinib (1 μM) and/or TNO155 (1 μM) for 24 h. Cle-PARP, cleaved poly (ADP-ribose) polymerase.

## Data Availability

The data generated are available upon reasonable request from the corresponding author.

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
