# Peer review of "Diverse Mechanisms of Resistance against Osimertinib, a Third-Generation EGFR-TKI, in Lung Adenocarcinoma Cells with an EGFR-Activating Mutation"

_cells, 2022, doi:10.3390/cells11142201_

Round 1

Reviewer 1 Report

The manuscript "cells-1744166" by Shigetoshi Nishihara et al. "Diverse Mechanisms of Resistance against Osimertinib, a Third Generation EGFR-TKI, in Lung Adenocarcinoma Cells with an EGFR-activating Mutation" is well written and easy to follow . It provides a "proof of concept" that multiple mechanisms can lead to Osimertinib resistance, underlining the importance of identify the specific mechanism of resistance for each EGFR-patient.   

Minor comments:

1) Conclusion should be extended. May I suggest to strength the importance of identify resistance mechanisms to Osimertinib.

2) What is the level of c-MET in the clone AZDR14?

Nonetheless I would recommend it for publication.

Author Response

Responses to the comments of Reviewer #1:

The manuscript "cells-1744166" by Shigetoshi Nishihara et al. "Diverse Mechanisms of Resistance against Osimertinib, a Third Generation EGFR-TKI, in Lung Adenocarcinoma Cells with an EGFR-activating Mutation" is well written and easy to follow. It provides a "proof of concept" that multiple mechanisms can lead to Osimertinib resistance, underlining the importance of identify the specific mechanism of resistance for each EGFR-patient.   

Response:

We thank you for your positive evaluation regarding our manuscript.

Minor comments:

1) Conclusion should be extended. May I suggest to strength the importance of identify resistance mechanisms to Osimertinib.

Response:

We agree with your comment. We have revised the Conclusion to include the importance of identifying resistance mechanisms to osimertinib (in yellow highlight) (Lines 491–496).

2) What is the level of c-MET in the clone AZDR14?

Response:

We thank you for your insightful comment. We have provided the western blot images of c-MET in Figure 1c, which show that c-MET expression levels were similar in parental PC-9 and AZDR14 cells.

Reviewer 2 Report

The authors analyse a single non-small cell cancer cell line by treating the NSCLC cell line PC-9 with the drug Osimertinib. Analyses of five ensuing resistant derivatives of this line show heterogeneous mechanisms of resistance have arisen. The aim is to provide mechanistic insights to develop novel therapeutic strategies for Osimertinib resistant NSCLC.

Comments:

1) Only a single NSCLC cell line is resistance selected. While a heterogeneous set of resistant cells reveal multiple resistance mechanisms, the authors throughout generalise their findings to NSCLC. This is inappropriate a only the PC-9 cell line was studied.

2) Clinically osimeritinib is delivered in combination with pemetrexed or cisplatin. When investigating potential clinical resistance mechanisms those which arise from combination resistant are most relevant. This is not presented in this study.

3) Xenograft experiments are completed in fully immunocompromised mice thus any alterations arising from immune of TME are absent in this tissue culture based study.

From these three points throughout the manuscript references to wider study of NSCLC and clinical applicability to NSCLC should be removed. 

4) A number of mechanisms discovered have been individually reported elsewhere, such as the use of TNO155 in SHP2/T507K osimertinib resistance. 

5) Keita Masuzawa et al (10.18632/oncotarget.22297) state that  “For classic EGFR mutations (exon 19 deletion and L858R, with or without T790M), osimertinib showed lower IC50 values and wider therapeutic windows than nazartinib. For less common EGFR mutations (G719S or L861Q), afatinib showed the lowest IC50 values. For G719S+T790M or L861Q+T790M, the IC50 values of osimertinib and nazartinib were around 100 nM, which was 10- to 100-fold higher than those for classic+T790M mutations.” The methodology in the submitted manuscript describes a dose-escalation strategy of gradual Osimertinib dose escalation. Is this appropriate? Given the IC50 dependencies did this strategy of forming resistant cells then force the cells along certain pathways of resistance that would not necessarily be observed in patients treated at clinical dose?

6) Methods 2.6 Ras pull-down assay does not contain sufficient methodology description. 

7) Why were only female mice used? 

8) Statistical analysis is not appropriate. In the murine xenografts data should be presented in supplementary material showing each individual mouse curve. Is a single student t-test with p<0.05 appropriate for every experiment in the paper? Would ANOVA be more suitable for tumour xenograft experiments? Its surprising that every methodology presented in the manuscript can all be analysed via a single statistical method and p value? 

9) Why was the AZDR3 line tested in vivo as a xenograft and not other resistant lines? 

10) In Figure 3 descriptions of (e) and (f) in Figure legend have concentration until missing. 

11) Western images supplied as original images are 1) saturated in a number of cases beyond quantitation and 2) appear to be different exposures of the same experiment in some cases as opposed to repeats. Please supply the three similar levels of exposures from the experimental reproductions of the data. 

There are some clear technical and scientific issues with this manuscript, the Westerns supplied clearly have an issue. Please address issues throughout.

Author Response

Responses to the comments of Reviewer #2:

“The authors analyse a single non-small cell cancer cell line by treating the NSCLC cell line PC-9 with the drug Osimertinib. Analyses of five ensuing resistant derivatives of this line show heterogeneous mechanisms of resistance have arisen. The aim is to provide mechanistic insights to develop novel therapeutic strategies for Osimertinib resistant NSCLC.”

Response:

We thank you for your evaluation regarding our manuscript.

Comments:

1) Only a single NSCLC cell line is resistance selected. While a heterogeneous set of resistant cells reveal multiple resistance mechanisms, the authors throughout generalise their findings to NSCLC. This is inappropriate a only the PC-9 cell line was studied.

Response:

We thank you for pointing this out and we agree that our findings need to be generalized using other NSCLC cell lines with EGFR activating mutation. As mentioned in the discussion, the resistance mechanisms are greatly heterogenous; therefore, it would be difficult to prepare cells carrying the same resistance mechanisms. Clinically, the resistance mechanisms to osimertinib are vastly heterogeneous, and in approximately half of the patients, resistance mechanisms to osimertinib cannot be determined for first line therapy (British Journal of Cancer 2019, 121:725–737 and Nature Cancer 2021, 2:377-391). Therefore, identifying novel resistance mechanisms to osimertinib as first line therapy for improving cancer therapy is required.

2) Clinically osimeritinib is delivered in combination with pemetrexed or cisplatin. When investigating potential clinical resistance mechanisms those which arise from combination resistant are most relevant. This is not presented in this study.

Response:

           We thank you for the valuable comment. Following FDA approved therapy and Japanese therapeutical guidelines, the combination therapy with osimertinib is not approved. In 2018, osimertinib was approved for the first-line treatment of patients with metastatic NSCLC whose tumors have EGFR mutations (exon 19 deletions or exon 21 L858R mutations). The approval was based on results from the Phase III FLAURA trial, which were published in the New England Journal of Medicine (NEJM 2018, 378, 113-125). Therefore, in this study, PC-9 cells have been treated with osimertinib as first-line therapy, and the resistant cells were established.

3) Xenograft experiments are completed in fully immunocompromised mice thus any alterations arising from immune of TME are absent in this tissue culture based study. From these three points throughout the manuscript references to wider study of NSCLC and clinical applicability to NSCLC should be removed. 

Response:

Thank you for your insightful comment. We agree that references to wider study of NSCLC and clinical applicability to NSCLC should be removed. Accordingly, we have checked our manuscript and removed any such expressions.

4) A number of mechanisms discovered have been individually reported elsewhere, such as the use of TNO155 in SHP2/T507K osimertinib resistance. 

Response:

We thank you for the thoughtful and helpful comment. To the best of our knowledge, there is no report on the use of TNO155 in osimertinib-resistant mutation of SHP2/T507K. SHP2/T507K mutation in PTPN11 of exon13 was reported in patients with neuroblastoma (Cancer Research 64, 8816-8820, 2004). It has been reported that the PTPN11 alteration in NSCLC is 0.93% (Cancer Discovery. 2017;7(8):818-831. Dataset Version 8). Furthermore, there is no report that PTPN11/T507K was identified in NSCLC. Therefore, we believe our findings valuable to be published.

5) Keita Masuzawa et al (10.18632/oncotarget.22297) state that “For classic EGFR mutations (exon 19 deletion and L858R, with or without T790M), osimertinib showed lower IC50 values and wider therapeutic windows than nazartinib. For less common EGFR mutations (G719S or L861Q), afatinib showed the lowest IC50 values. For G719S+T790M or L861Q+T790M, the IC50 values of osimertinib and nazartinib were around 100 nM, which was 10- to 100-fold higher than those for classic+T790M mutations.” The methodology in the submitted manuscript describes a dose-escalation strategy of gradual Osimertinib dose escalation. Is this appropriate? Given the IC50 dependencies did this strategy of forming resistant cells then force the cells along certain pathways of resistance that would not necessarily be observed in patients treated at clinical dose?

Response:

We thank you for the thoughtful and relevant comment. We agree with Reviewer #2’s concern about the methodology to establish resistance cell lines and the appropriate concentration of osimertinib for clinical dose. Pharmacokinetic studies of osimertinib have revealed that the maximum plasma concentration (Cmax) of osimertinib was 381–485 nM (Br J Clin Pharmacol. 2017; 83(6): 1216–1226) at the approved dose of osimertinib 80 mg, daily. In the interview form from Japanese pharmaceuticals and medical devices agency, the Cmax of osimertinib following multiple oral doses of osimertinib 80 mg daily to Japanese NSCLC patients was 0.78 μM. Therefore, if osimertinib-resistant cells are developed using the increasing dose escalation method, the final concentration of osimertinib, 1.0 μM, should be appropriate and clinically relevant.

6) Methods 2.6 Ras pull-down assay does not contain sufficient methodology description. 

Response:

We apologize for the unclear description in Method 2.6. We have now added detailed information to explain the Ras pull-down assays in the manuscript. The changes are highlighted in yellow (lines 122–124).

7) Why were only female mice used? 

Response:

We thank you for your insightful comment. We employed female SCID mice in xenograft studies on tumor growth. Female mice are easier to handle compared to male mice because male mice fight over territory. Due to more aggression among male mice, female mice are frequently preferred for animal experiments.

8) Statistical analysis is not appropriate. In the murine xenografts data should be presented in supplementary material showing each individual mouse curve. Is a single student t-test with p<0.05 appropriate for every experiment in the paper? Would ANOVA be more suitable for tumour xenograft experiments? Its surprising that every methodology presented in the manuscript can all be analysed via a single statistical method and p value? 

Response:

We thank the reviewer for raising this important point regarding statistical analysis and murine xenograft data. Generally, t-test and Analysis of Variance abbreviated (ANOVA) are two parametric statistical techniques used to test the hypothesis. t-test is a hypothesis test that is used to compare the means of two populations. ANOVA is a statistical technique that is used to compare the means of more than two populations. In this study, we compared the means of two populations, therefore, t-test was employed. Furthermore, we apologize for the oversight in marking “*” in Figure 2f; we have now modified Figure 2f to include this.

9) Why was the AZDR3 line tested in vivo as a xenograft and not other resistant lines? 

Response:

We thank the reviewer for providing this helpful suggestion. We have tested using the xenograft mouse model of AZDR3, AZDR6 and AZDR6 F2M cells, as given in Figures 2f and S3g. In contrast to an in vitro study, in vivo studies using xenograft mice require large amounts of inhibitors. We can obtain osimertinib and afatinib in large quantities from the pharmaceutical company; however, obtaining other inhibitors, such as linsitinib, selumetinib, and TNO155, would be difficult. If ever it is feasible to perform xenograft studies using these inhibitors, we will certainly take up those studies.

10) In Figure 3 descriptions of (e) and (f) in Figure legend have concentration until missing. 

Response:

We apologize for the oversight. We have made appropriate revisions in Figure 3e and f. The changes are highlighted in yellow (lines 275–277).

11) Western images supplied as original images are 1) saturated in a number of cases beyond quantitation and 2) appear to be different exposures of the same experiment in some cases as opposed to repeats. Please supply the three similar levels of exposures from the experimental reproductions of the data. There are some clear technical and scientific issues with this manuscript, the Westerns supplied clearly have an issue. Please address issues throughout.

Response:

We agree with the reviewer’s recommendation to “supply the three similar levels of exposures from the experimental reproductions of the data”. Therefore, we have provided three images of similar exposure levels.

Thank you again for considering publication of our revised report in Cells. We believe that addressing the Editors and Reviewers’ comments has substantially improved the quality and impact of the manuscript. Please do not hesitate to contact me if you have further questions or clarification.

Reviewer 3 Report

This is an interesting manuscript, but a few points need to be addressed before it can be considered acceptable for publication:

1. The statement in lines 55/56 "Therefore, osimertinib is recommended as first-line treatment for patients with EGFR-mutant NSCLC" is incorrect. Osimertinib is only recommended for certain EGFR-mutations, including exon-19 deletions. 

2. please make it clear that PC-9 represents NSCLC that would be treated with front line osimertinib.

3. Where the osimertinib-resistant clones profiled by STR sequencing at the end of the selection process. This is essential considering that the cell line s were cultured for over 12 months.

4. The authors need to acknowledge that these experiments were performed in a single cell line (PC-9 and its resistant derivatives), notably that this work is unable to account for cell-line specific effects.

5. In section 3.2 - a further validation experiment using siRNA targeting the region of exon 19 that is deleted in PC-9 would be highly interesting/relevant.  By specifically targeting wild-type EGFR in AZDR3 cells, would one restore sensitivity to osimertinib?

6. Are T507K PTPN11 mutations observed in NSCLC patients?

Author Response

Nishihara et al. “Diverse Mechanisms of Resistance against Osimertinib, a Third-generation EGFR-TKI, in Lung Adenocarcinoma Cells with an EGFR-activating Mutation” (manuscript ID: Cells-1744166)

Responses to the comments of Reviewer #3:

This is an interesting manuscript, but a few points need to be addressed before it can be considered acceptable for publication:

Response

We thank Reviewer #3 for his/her overall evaluation that “This is an interesting manuscript”.

Comments:

1. The statement in lines 55/56 "Therefore, osimertinib is recommended as first-line treatment for patients with EGFR-mutant NSCLC" is incorrect. Osimertinib is only recommended for certain EGFR-mutations, including exon-19 deletions. 

Response

We appreciate the reviewer raising this important point. We agree that EGFR mutation genotypes, such as exon19 deletion and L858R, or human races, affect the efficacy of osimertinib. In the FLAURA study, in comparison to first-generation EGFR-TKIs (gefitinib or erlotinib), osimertinib has substantially longer median of progression-free survival (18.9 months versus 10.2 months) with lower toxicity profile. This result leads to approval as a standard first-line treatment for EGFR-mutated NSCLC. However, the survival sub-group analysis in this trial indicated that Asian and the L858R-mutated populations did not show a benefit over first-generation EGFR-TKIs. Therefore, we have revised the sentences to “osimertinib is recognized as first-line treatment for patients with EGFR-mutant NSCLC.” The changes are highlighted in yellow (lines 55–57).

2. please make it clear that PC-9 represents NSCLC that would be treated with front line osimertinib.

Response

We thank the reviewer’s thoughtful and relevant comment. We agree to emphasize that PC-9 cells were established from untreated patients, and these established osimertinib-resistant cell lines were from PC-9 cells as first-line therapy. The changes are highlighted in yellow (lines 88–90).

3. Where the osimertinib-resistant clones profiled by STR sequencing at the end of the selection process. This is essential considering that the cell lines were cultured for over 12 months.

Response

We agree with the reviewer’s comment that it is necessary to include information regarding cell line authentication of these osimertinib-resistant cell lines for excluding cross-contamination of cell lines. We have received cell line authentication at the Japanese Collection of Research Bioresources Cell Bank and have added this sentence in Materials and Method section. The changes are highlighted in yellow (lines 81–83).

4. The authors need to acknowledge that these experiments were performed in a single cell line (PC-9 and its resistant derivatives), notably that this work is unable to account for cell-line specific effects.

Response

We appreciate the reviewer raising this important point. We agree the need to generalize these findings using the other NSCLC cell lines with EGFR activating mutation. As we mentioned in the discussion, the resistance mechanisms are greatly heterogenous; therefore, it would be difficult to prepare cells carrying the same resistance mechanisms. Clinically, the resistance mechanisms to osimertinib have been vastly heterogeneous, and in approximately half of the patients, resistance mechanisms to osimertinib as first line therapy cannot be determined (British Journal of Cancer 2019, 121:725–737 and Nature Cancer 2021, 2:377-391). Therefore, it is required to identify novel resistance mechanisms to osimertinib as first line therapy for improving cancer therapy. Therefore, in the next step, we would like to find same resistance mechanisms in human tissues, which have acquired resistance to osimertinib. The limitation in this study was added in the conclusion section (lines 489–491). The changes are highlighted in yellow.

5. In section 3.2 - a further validation experiment using siRNA targeting the region of exon 19 that is deleted in PC-9 would be highly interesting/relevant.  By specifically targeting wild-type EGFR in AZDR3 cells, would one restore sensitivity to osimertinib?

Response

        We thank reviewer for his/her helpful suggestion. In section 3.2, we have employed siRNA against human EGFR (Dharmacon), targeting either wild-type and the region of EGFR exon 19. This siRNA should attenuate whole EGFR including mutated EGFR and wild type EGFR. Therefore, the cells, whose major survival signal is dependent on EGFR (including wild-type and mutated EGFR), show reduced cell proliferation and the induction of apoptosis is enhanced by the attenuation of EGFR. As shown in Figure 3e, PC-9 and AZDR3 cells exhibited remarkable apoptosis induction by siRNA against EGFR transfection. Moreover, as shown below, cell-proliferation of PC-9 and AZDR3 was decreased, when siRNA against EGFR was transfected in the right figure (Please use attached file).

6. Are T507K PTPN11 mutations observed in NSCLC patients?

Response

        To the best of our knowledge, we could not find any reports regarding PTPN11/T507K mutation in NSCLC patients. Therefore, ours is the first study to report that PTPN11/T507K occurred in NSCLC cell lines, and furthermore, that it confers resistance to osimertinib.

Thank you again for considering publication of our revised report in Cells. We believe that addressing the Editors and Reviewers’ comments has substantially improved the quality and impact of the manuscript. Please do not hesitate to contact me if you have further questions or clarification.

Reviewer 4 Report

In this article, the authors have been reported and explored acquired resistance mechanisms (ARM) to Osimertinib. These ARM have been studied in cell lines harbouring the most common EGFR mutation (exon 19 del) and exposed to low dose of Osimertinib to generate cell lines with ARM to Osimertinib. They have been shown that these ARM encompass EGFR-dependent (EGFR amplification) as well as EGFR independent mechanisms (bypass signaling, downstream pathway activation including MAPK activation, bypass signal of IGF1). They also reported that Bid cleavage by caspase 9 is required in cell with bypassing signal of IGF1 as an ARM.

I have some comments:

My main concern is the fact that ARMs reported in this work are not the must common ARM found in patients exposed to Osimertinib in both 1st  and  2nd line setting (see  N. Roper et al., Cell Reports Medicine 1, (2020)) and A Leonetti et al. British Journal of Cancer (2019) 121:725–737). How does the authors could explain or discuss this issue in discussion part?

Line 58: “and data on the underlying mechanisms are limited”, line 61-62 “Nevertheless, approximately half of the resistance-related mechanisms remain unknown”:  ARMs to Osimertinib in both 1st and 2nd setting are well reported and described by roper (N. Roper et al., Cell Reports Medicine 1, (2020)) and Leonetti (ref 8 in this paper). Therefore, the authors cannot claim that data are limited. Additionally, the availability of liquid biopsy as well the option of targeted therapy such as the combination of the EGFR–MET bispecific antibody amivantamab with the third-generation EGFR TKI lazertinib (chrysalis trial) in patient with ARM to Osimertinib favour rebiopsy in this context as sshown by Roper (N. Roper et al., Cell Reports Medicine 1, (2020). Thus, the assertion need  to be mitigate.

It’s could be great if the authors could evaluate the combination of Osimertinib and Necitumumab ( a humanized IgG1 anti-EGFR) in AZDR3 harbouring EGFR amplification as an ARM. This could be interesting because Necitumumab has been evaluated and shown efficacy in squamous NSCLC.

Line  402: mutation instead of “muatation”

Author Response

Responses to the comments of Reviewer #4:

In this article, the authors have been reported and explored acquired resistance mechanisms (ARM) to Osimertinib. These ARM have been studied in cell lines harbouring the most common EGFR mutation (exon 19 del) and exposed to low dose of Osimertinib to generate cell lines with ARM to Osimertinib. They have been shown that these ARM encompass EGFR-dependent (EGFR amplification) as well as EGFR independent mechanisms (bypass signaling, downstream pathway activation including MAPK activation, bypass signal of IGF1). They also reported that Bid cleavage by caspase 9 is required in cell with bypassing signal of IGF1 as an ARM.

Comments:

1. My main concern is the fact that ARMs reported in this work are not the most common ARM found in patients exposed to Osimertinib in both 1stand  2ndline setting (see  N. Roper et al., Cell Reports Medicine 1, (2020)) and A Leonetti et al. British Journal of Cancer (2019) 121:725–737). How does the authors could explain or discuss this issue in discussion part?

Response

              We appreciate the reviewer raising this important point. Regarding the previous reports of Roper et al, Leonetti et al, and Passaro et al (Nature Cancer 2021,2:377-391), the resistance mechanisms to osimertinib are vastly heterogenous. It is difficult to determine the most common acquired resistance mechanisms to osimertinib, such as T790M EGFR mutation, which is found in 50–60% of patients receiving gefitinib, erlotinib, or afatinib. In the article of Roper et al, they indicated that “MET amplification occurs in 66% (n=6/9) of first-line osimertinib-treated patients, albeit spatially heterogeneous, often co-occurs with additional acquired focal copy-number amplifications and is associated with early progression” and “The most common focal copy-number amplification emerged in the EGFR gene in post-osimertinib-resistance patients (n=5)”. In the articles of Leonetti et al, or Passaro et al, acquired EGFR mutations, EGFR amplification, and MET amplification might be relatively major resistance mechanisms to osimertinib, although all of the resistance mechanisms could not be clarified. Interestingly, in our established osimertinib-resistant cells, AZDR3, AZDR6, and AZDR11, EGFR amplification is substantial, as shown in Figure 1d. In addition to EGFR gene amplification, MET was overexpressed at the protein level in AZDR3 cells (Figure 1c), and KRAS gene amplification was observed in AZDR6 cells (Figure 3a). Therefore, these cell lines have multiple acquired focal copy number amplifications. Unfortunately, acquired EGFR mutation could not be identified in our resistant cells to osimertinib. Therefore, we should acknowledge that these resistant cells could not cover all of the resistance mechanisms to osimertinib. We have added sentences in the conclusion section (lines 489–491). The changes are highlighted in yellow.

 2. Line 58: “and data on the underlying mechanisms are limited”, line 61-62 “Nevertheless, approximately half of the resistance-related mechanisms remain unknown”:  ARMs to Osimertinib in both 1st and 2nd setting are well reported and described by roper (N. Roper et al., Cell Reports Medicine 1, (2020)) and Leonetti (ref 8 in this paper). Therefore, the authors cannot claim that data are limited. Additionally, the availability of liquid biopsy as well the option of targeted therapy such as the combination of the EGFR–MET bispecific antibody amivantamab with the third-generation EGFR TKI lazertinib (chrysalis trial) in patient with ARM to Osimertinib favour rebiopsy in this context as shown by Roper (N. Roper et al., Cell Reports Medicine 1, (2020). Thus, the assertion need to be mitigate.

Response

              We agree with the reviewer’s recommendation. With the great efforts being put in to discover the resistance mechanisms to osimertinib, the novel resistance mechanisms are accumulating. Therefore, we have revised the expressions in these sentences (lines 59– 60 and lines 63–64). The changes are highlighted in yellow.

 3. It’s could be great if the authors could evaluate the combination of Osimertinib and Necitumumab ( a humanized IgG1 anti-EGFR) in AZDR3 harbouring EGFR amplification as an ARM. This could be interesting because Necitumumab has been evaluated and shown efficacy in squamous NSCLC.

Response

              We agree with the reviewer’s comment that the combination effect of osimertinib and necitumumab in AZDR3 cells could be evaluated. Unfortunately, we could not obtain necitumumab for this study. However, instead of necitumumab, cetuximab, which is an anti-EGFR recombinant monoclonal antibody, was tested. As shown below, its suppressive effect on cell proliferation was modest, when cetuximab was exposed to AZDR3 cells, either in the presence or absence of osimertinib in the below left figure. Moreover, EGFR phosphorylation was inhibited modestly by cetuximab treatment. Cetuximab inhibited AKT activation, but not ERK1/2 activation. The apoptosis induction was slightly observed in cetuximab treatment, and the induction was slightly enhanced in the presence of osimertinib. We agree that these data should be premature; however, we suggested that afatinib exhibited higher efficacy for the wild-type EGFR amplified osimertinib-resistant AZDR3 cells than the combination of osimertinib and cetuximab. (Figures were unable to be uploaded, Please use an attached file.)

4. Line  402: mutation instead of “muatation”

Response

              We apologize for this error. We have corrected this mistake in the revised manuscript. This change is highlighted in yellow (line 382).

Thank you again for considering publication of our revised report in Cells. We believe that addressing the Editors and Reviewers’ comments has substantially improved the quality and impact of the manuscript. Please do not hesitate to contact me if you have further questions or clarification.

Reviewer 5 Report

The manuscript ‘Diverse Mechanisms of Resistance against Osimertinib, a Third Generation EGFR-TKI, in Lung Adenocarcinoma Cells with an EGFR-activating Mutation’ is an interesting finding on basis of the resistance mechanism. This study gives vital information to the clinic.  However, few control experiments need to be done.

Comments:

1.    Establishment of the acquired osimertinib-resistant PC-9 cells method needs to be detailed. Were the resistance cells acquired from single-cell clones or? Also, how are the resistance cells maintained?

2.    ‘selumetinib inhibited both AKT and ERK1/2 phosphorylation in a dose-dependent manner (Figure 3f)’. MEK inhibitor reduces AKT phosphorylation does it because selumetinib is off-target or? The authors need to comment on this. Selumetinib spelling is wrong in figure 3f. Do other FDA-approved MEK inhibitors show the same effect?.

3.    Does the combination in figures 3c and d, figure 4b show a synergistic or additive effect?

4.    Figure6, high concentration of three drugs reduce cell viability and downstream signaling does this indicates toxicity in these cells?

5.    Since SHP2/T507K with Gab1 hyperphosphorylates RAS does MEK inhibitor sensitize the AZDR14 cells?

Author Response

Responses to the comments of Reviewer #5:

The manuscript ‘Diverse Mechanisms of Resistance against Osimertinib, a Third Generation EGFR-TKI, in Lung Adenocarcinoma Cells with an EGFR-activating Mutation’ is an interesting finding on basis of the resistance mechanism. This study gives vital information to the clinic.  However, few control experiments need to be done.

Response

              We appreciate the reviewer’s overall evaluation of our study that it “is an interesting finding,” and we are pleased that the reviewer recognized that “This study gives vital information to the clinic.

Comments:

1. Establishment of the acquired osimertinib-resistant PC-9 cells method needs to be detailed. Were the resistance cells acquired from single-cell clones or? Also, how are the resistance cells maintained?

Response

We agree with the reviewer on the need to provide further commentary regarding establishment and maintenance of osimertinib-resistant cell lines from PC-9 cells. All osimertinib-resistant cells did not receive single-cell cloning. Instead of single cell cloning, we have maintained them by culturing in the growth medium containing 1 mM osimertinib for at least 2–3 months. These osimertinib-resistant cells were maintained continuously in the growth medium containing 1 mM osimertinib. We have provided the sentences in Materials & Methods section (lines 93–95). The changes are highlighted in yellow.

2. ‘selumetinib inhibited both AKT and ERK1/2 phosphorylation in a dose-dependent manner (Figure 3f)’. MEK inhibitor reduces AKT phosphorylation does it because selumetinib is off-target or? The authors need to comment on this. Selumetinib spelling is wrong in figure 3f. Do other FDA-approved MEK inhibitors show the same effect?

Response

              We appreciate the reviewer for the careful review of our data and agree with the reviewer’s comment that “MEK inhibitor reduces AKT phosphorylation does it because selumetinib is off-target or?”. It is important to elucidate whether the precise mechanisms of MEK inhibitor reduces AKT phosphorylation. As the reviewer has pointed out, we have tried to use trametinib, which is also an FDA-approved MEK inhibitor on AZDR9 cells. As shown below, in AZDR9, which expresses KRASG13D mutation in addition to EGFR exon 19 deletion, AKT and ERK1/2 phosphorylation were reduced after treatment with trametinib. Otherwise, osimertinib did not affect AZDR9 cells. Similar to selumetinib, trametinib also reduced AKT and ERK1/2 activation. Therefore, it suggests that it might not be an off-target effect. Others reported that acquired KRASG12V mutation in PC-9 cells showed resistance to osimertinib (Cancer Science 2021;112:3784–3795). In this article, trametinib inhibited ERK1/2 activation, but not AKT activation. Moreover, we could agree this further investigation could strengthen the mechanistic understanding of resistance in the future studies. We have added sentence regarding this effect (lines 249– 255). We have added this figure as Figure S3a. (The Figures were unable to be uploaded, please use an attached file.)

              We apologize for this error. We have made the appropriate revisions to Figure 3f.

3. Does the combination in figures 3c and d, figure 4b show a synergistic or additive effect?

Response

              We agree with the reviewer on the need to provide the combination effect of to afatinib and selumetinib in AZDR6 cells (Figure 3c) and of osimertinib and linsitinib in AZDR11 cells (Figure 4b). Afatinib or selumetinib alone did not inhibit cell proliferation in AZDR6 cells; thus, combination of afatinib and selumetinib should have a synergistic effect of suppression in AZDR6 cells. Osimertinib or linsitinib alone did not inhibit cell-proliferation in AZDR11 cells, thus it should be synergistic suppression of AZDR11 cells by combination of osimertinib and linsitinib.

4. Figure6, high concentration of three drugs reduce cell viability and downstream signaling does this indicates toxicity in these cells?

Response

              As pointed out by the reviewer, it would be meaningful to consider cell toxicity with the exposure of combination of multiple inhibitors, such as osimertinib, linsitinib, and TNO155, which is an EGFR-TKI, IGF1R inhibitor, and SHP2 inhibitor, respectively. If the combination of these inhibitors affected AZDR14 cells as toxic, non-specific inhibition of signals should be exhibited. In Figure 6f, TNO155, SHP2 inhibitor, alone inhibited GAB1 activation, which is the downstream effector of SHP2. The combination of osimertinib and linsitinib inhibited EGFR and IGF1R phosphorylation and downstream of AKT activation, this effect agreed with the effect of Figure 6c. Moreover, when TNO155 was exposed dose-dependently in addition with the combination of osimertinib 1 mM and linsitinib 1 mM in Figure 6f, TNO155 inhibited GAB1 phosphorylation dose-dependently. Furthermore, EGFR and IGF1R phosphorylation were not affected by TNO155. Therefore, non-specific effect could be excluded.

5. Since SHP2/T507K with Gab1 hyperphosphorylates RAS does MEK inhibitor sensitize the AZDR14 cells?

Response

              We appreciate the reviewer raising this important point of clarification regarding MEK inhibition in RAS activated SHP2/K507K osimertinib-resistant cells. Based on the need to evaluate the effect of MEK inhibitor in cell-proliferation and signal transduction in AZDR14 cells, when AZDR14 cells were exposed to trametinib dose-dependently, ERK1/2 phosphorylation was clearly inhibited, and cell proliferation was partially suppressed. Although AKT inhibition did not occur, osimertinib was added to trametinib as shown in the lower figure.(The Figures were unable to be uploaded, please use an attached file.)

As shown in lower left figure, the activation of AKT was decreased using an IGF1R inhibitor. Therefore, as shown in lower right figure, under osimertinib and linsitinib exposure, trametinib inhibited ERK1/2 phosphorylation dose-dependently, leading to apoptosis induction.(The Figures were unable to be uploaded, please use an attached file.)

Furthermore, cell proliferation was greatly suppressed when trametinib was treated with a combination of osimertinib and linsitinib as shown in the lower figure. (The Figures were unable to be uploaded, please use an attached file.)

As indicated in these figures, trametinib, a MEK inhibitor, could sensitize AZDR14 cells under osimertinib and linsitinib treatment. We can agree with our data in Figure 6 that SHP2/T507K mutation confers hyper-phosphorylation RAS through GAB1.

Thank you again for considering publication of our revised report in Cells. We believe that addressing the Editors and Reviewers’ comments has substantially improved the quality and impact of the manuscript. Please do not hesitate to contact me if you have further questions or clarification.

Round 2

Reviewer 2 Report

The replacement of the Western images on file is appreciated. They really are very over-exposed, have no MW indications and have a number of positive lanes not labelled or identified. Given the number of alterations now made in this manuscript it is suitable for publication after editing checks. The previous comments regarding breadth of applicability, cell lines etc have not been addressed but the limitations of the work is now clearer within the manuscript. 

Author Response

Responses to the comments of Reviewer #2:
“The replacement of the Western images on file is appreciated. They really are very over-exposed, have no MW indications and have a number of positive lanes not labelled or identified. Given the number of alterations now made in this manuscript it is suitable for publication after editing checks. The previous comments regarding breadth of applicability, cell lines etc have not been addressed but the limitations of the work is now clearer within the manuscript.”

Response:
We thank Reviewer #2 for the overall evaluation that the manuscript “is suitable for publication.”
Per your suggestion, molecular weight markers have been added to the original blot images. We understand the Reviewer's concern that the images “have a number of positive lanes not labelled or identified,” because we included samples to test some antibodies that could have identified novel or promising signaling molecules but were ultimately not used in this manuscript. Therefore, we could not label or identify the corresponding lanes. We appreciate your understanding.
Furthermore, English language and style in the manuscript have been checked by the professional English-editing service providing company, Editage.

Reviewer 3 Report

None

Author Response

Responses to the comments of Reviewer #3:
None of comments and suggestions for authors
Response
We thank Reviewer #3 for his/her overall evaluation regarding our manuscript.
For the publication of this manuscript, English language and style have checked by English-editing service, Editage.
